# Carrot DcALFIN4 and DcALFIN7 Transcription Factors Boost Carotenoid Levels and Participate Differentially in Salt Stress Tolerance When Expressed in *Arabidopsis thaliana* and *Actinidia deliciosa*

**DOI:** 10.3390/ijms232012157

**Published:** 2022-10-12

**Authors:** Luis Felipe Quiroz-Iturra, Kevin Simpson, Daniela Arias, Cristóbal Silva, Christian González-Calquin, Leticia Amaza, Michael Handford, Claudia Stange

**Affiliations:** 1Genetics & Biotechnology Lab, Plant & AgriBiosciences Research Centre (PABC), Ryan Institute, University of Galway, University Road, H91 REW4 Galway, Ireland; 2Departamento de Genética Molecular y Microbiología, Facultad de Ciencias Biológicas, Pontificia Universidad Católica de Chile, Alameda 340, Santiago 7750000, Chile; 3Departamento de Biología, Facultad de Ciencias, Universidad de Chile, Las Palmeras 3425, Ñuñoa, Santiago 7750000, Chile

**Keywords:** ALFIN-like transcription factor, DcAL4 and DcAL7, carrot, carotenoids, salt stress tolerance, *Actinidia deliciosa* (kiwi)

## Abstract

ALFIN-like transcription factors (ALs) are involved in several physiological processes such as seed germination, root development and abiotic stress responses in plants. In carrot (*Daucus carota*), the expression of *DcPSY2*, a gene encoding phytoene synthase required for carotenoid biosynthesis, is induced after salt and abscisic acid (ABA) treatment. Interestingly, the *DcPSY2* promoter contains multiple ALFIN response elements. By in silico analysis, we identified two putative genes with the molecular characteristics of ALs, *DcAL4* and *DcAL7*, in the carrot transcriptome. These genes encode nuclear proteins that transactivate reporter genes and bind to the carrot *DcPSY2* promoter in yeast. The expression of both genes is induced in carrot under salt stress, especially *DcAL4* which also responds to ABA treatment. Transgenic homozygous T3 *Arabidopsis thaliana* lines that stably express *DcAL4* and *DcAL7* show a higher survival rate with respect to control plants after chronic salt stress. Of note is that *DcAL4* lines present a better performance in salt treatments, correlating with the expression level of *DcAL4*, *AtPSY* and *AtDXR* and an increase in carotenoid and chlorophyll contents. Likewise, *DcAL4* transgenic kiwi (*Actinidia deliciosa*) lines show increased carotenoid and chlorophyll content and higher survival rate compared to control plants after chronic salt treatment. Therefore, *DcAL4* and *DcAL7* encode functional transcription factors, while ectopic expression of *DcAL4* provides increased tolerance to salinity in Arabidopsis and Kiwi plants.

## 1. Introduction

The ALFIN-Like transcription factor (ALs) belongs to a small plant-specific subfamily of PHD (plant homeodomain) finger proteins [1]. All ALs possess three domains: the conserved C-terminal PHD domain (Cys4HisCys3), the conserved N-terminal PAL (PHD-associated ALFIN) domain and the V domain (variable region between PAL and PHD domains) [2,3]. The PHD domain was discovered in the HAT3.1 protein (Histone acetyltransferase 3.1) [4] with similarly to the RING-finger domain (Cys3HisCys4) that binds to two zinc atoms [5]. Most ALs recognizes the active histone markers (H3K4me3/2), participating in the chromatin remodeling process [3,6]. Exceptionally, *A. thaliana* AtAL3 lacks the conserved Tyr residue within the PHD domain (Figure 1a) preventing its binding to H3K4me3/2 [6]. On the other hand, the PAL domain (only found in ALs), recognizes the conserved G-Rich cis element GNGGTG/GTGGNG located in promoters of target genes and interacts with other transcription factors and components of the epigenetic regulatory machinery [2,3,7].

ALs are involved in multiple processes, such as root development, seed germination, root hair elongation, meristem development and abiotic stress tolerance [2,9,10,11]. For example, alfalfa (*Medicago sativa*) calli and plants that overexpress *MsAL1* are able to grow in high salt levels [7,12] and present an increase in root growth under normal conditions and salinity [9]. On the contrary, alfalfa calli expressing antisense *MsAL1* are more sensitive to salt treatments [12], indicating that *MsAL1* participates in salt stress tolerance. Among the seven *ALs* reported in *A. thaliana*, *AtAL5* has the highest transcriptional response under different abiotic stress conditions [2]. *AtAL5* mutants have reduced tolerance to high salt and drought, while plants that overexpress *AtAL5* have higher tolerance to these abiotic stresses [2]. In addition, heterologous expression of *AL1* from *Atriplex hortensis* (*AhAL1*) in *A. thaliana* enhances the abscisic acid (ABA) response and increases the survival rate and performance of transgenic plants under salinity and drought treatment [13].

One of the prime effects of abiotic stress is the production of radical oxygen species (ROS), affecting the photosynthesis and multiple cellular process [14,15,16,17]. Carotenoids are important molecules, which participate in photosynthesis and in protecting photosystem from the excess of light energy and photo oxidative damage [18,19,20]. In addition, they are powerful antioxidants and precursors of stress signaling molecules [21,22] such as ABA, an important hormone involved in the response to abiotic stresses [22,23,24]. Due its role during abiotic stress as antioxidants and ABA precursors, transgenic plants with increased carotenoid content possess higher tolerance to abiotic stresses [25,26,27,28,29,30].

Carrot (*Daucus carota*), one of the main vegetables consumed worldwide, synthesizes high levels of carotenoids in the taproot grown underground [31]. Storage root development takes around three months [32,33] and is accompanied by a substantial increase in size and carotenoid levels in correlation with an upregulation of the expression of most carotenogenic genes [29,31]. To date, several carotenogenic genes have been identified and functionally characterized in carrot and plant models [34,35,36,37,38,39]. In this regard, the constitutive expression of the lycopene β-cyclase 1 (*DcLCYB1*) in carrot produced a significant increase in total carotenoids in the taproot and in leaves of transgenic lines, showing its key role in carotenoid synthesis [34,36].

On the other hand, carrot is a moderately salt-tolerant species able to grow under 50–100 mM NaCl. Under acute salt stress treatment (250 mM NaCl), levels of carotenoid and ABA increase in leaves and root of carrot plants [8]. Phytoene synthase (PSY) is one of the key enzymes in carotenoid and ABA biosynthesis [40,41]. Carrot possesses two PSY (DcPSY1 and DcPSY2) [31] but it is *DcPSY2* which shows the highest increase in transcript levels under salt and ABA treatments [8]. In addition, the promoter of *DcPSY2*, but not that of *DcPSY1*, possesses ABA responsive elements (ABRE) that could be responsible for *DcPSY2* expression under ABA and salt stress [8], suggesting that ABA induces the expression of its own precursors. Nevertheless, other molecular players should be involved in the increased carotenoid content and induction of *DcPSY2* under stress condition.

In this work, we identified bioinformatically 11 putative ALs cis response elements present in the *DcPSY2* promoter and two putative ALs encoding genes in the carrot transcriptome, *DcAL4* and *DcAL7* (named due to their similarity to their orthologues in *A. thaliana*). Both *ALs* are differentially expressed in roots and leaves of carrot plants under salt and ABA treatment. Moreover, both ALs are localized in the nucleus, bind to the promoter of *DcPSY2* and induce the transactivation of reporter genes in yeast. By ectopic expression in *A. thaliana* and *Actinidia deliciosa* (kiwi), we determined that *DcAL4 A. thaliana* and *A. deliciosa* transgenic plants, but not *DcAL7* lines, show increased tolerance to salt stress and present an increase in chlorophyll and carotenoid levels. These results provide robust evidence that both *DcALs* encode functional nuclear transcription factors and that *DcAL4* promotes salt stress tolerance in plants.

## 2. Results

### 2.1. Bioinformatic Analysis of DcPSY2 Promoter and Identification of DcAL4 and DcAL7

Using PlantPAN 2.0 [42], PROMO [43,44] and PLACE [45], ALs cis binding motif were repeatedly found in the *DcPSY2* promoter (Figure 1a). Considering that it has been described that the ALs participate in the mechanisms of tolerance to abiotic stress [2,9,13], it was decided to study their possible role in the response to salt stress and *DcPSY2* induction.

Therefore, we searched for sequences with homology to MsAL1 in the carrot transcriptome [46] and the data base of plant transcription factors PlantTFDB [47]. After the alignment, the two putative amino acid sequences with the highest identity to MsAL1 were selected and named *DcAL4* and *DcAL7*. DcAL4 and DcAL7 possess 68.7% and 72,6% amino acid identity with *A. thaliana* AtAL4 and AtAL7, respectively. Using the NCBI CD-Search tool, it was corroborated that both sequences possess the PAL and PHD domains (Figure 1b), which are the main structural characteristics of ALs transcription factors. Specifically, they present the conserved amino acid sequences Cys4HisCys3 (canonical) and Tyr-Trp2 (necessary to bind H3K4me3/2) within the PHD domain (Figure 1b; highlighted in red and black, respectively). Therefore, the amino acid identity of DcAL4 and DcAL7 compared with their orthologues from *M. sativa* and *A. thaliana* suggests that *DcAL4* and *DcAL7* may encode functional transcription factors [3,12] involved in the gene regulation in response to abiotic stress [2,9,13].

### 2.2. DcAL4 and DcAL7 Encode for Functional Transcription Factors Able to Bind the DcPSY2 Promoter

For further characterization of *DcAL4* and *DcAL7*, we determined their subcellular localization by co-infiltration of tobacco leaves with 35SCaMV:GFP (as cytoplasm and nuclear marker) and DcAL4:RFP or DcAL7:RFP. The DcCAREB1:RFP construct was used as positive control for nuclear localization as *CAREB1* encodes a previously characterized bZIP transcription factor of *D. carota* [48]. Figure 2a shows that both *D. carota* recombinant ALs are localized in the nucleus, although DcAL4:RFP also shows a cytoplasmic localization. The nuclear localization of DcAL4 and DcAL7 was also confirmed through transfection of carrot protoplast (Appendix A).

Considering that 11 ALFIN responsive elements were found in 750 bp of the *DcPSY2* promoter (Figure 1a) and that in carrot the transcript levels of *DcPSY2* are induced under salt and ABA treatment [8], a yeast one-hybrid (Y1H) assay was performed to determine the ability of DcAL4 and DcAL7 to bind to the *DcPSY2* promoter. The *S. cerevisiae* Y1HGol-pP2-AbAi reporter strain transformed with vectors carrying *DcAL4* and *DcAL7* fused to the Gal4 AD (Activation Domain) survived in the presence of AbAi (Figure 2b), contrary to the empty vector pDEST22 and a vector carrying the *CAREB1* gene, which binds to motifs that are not present in the promoter of *DcPSY2*. This result suggests that, at least in yeast, DcAL4 and DcAL7 bind to the *DcPSY2* promoter.

To determine the DcALs capability to activate transcription, we accomplished a transactivation assay using the *S. cerevisiae* strain MaV203 [49]. Transformation of this strain with pDEST32 plasmid that carry *DcAL4* and *DcAL7* fused to the Gal4 DBD (DNA binding domain), allows the yeast to grow in a medium without uracil (SD/-Leu-Ura) and histidine (SD/-Leu-His supplemented with 25 and 50 mM 3AT) (Figure 2c), indicating that DcAL4 and DcAL7 induce the expression of the *URA3* and *HIS3* reporter genes. Yeast containing the empty pDEST32 plasmid were unable to grow in medium without uracil or histidine. CAREB1 transcription factor was used as a positive control, as it was proven previously to possess transactivation capability [48]. These results indicate that *DcAL4* and *DcAL7* encode for functional ALs.

### 2.3. DcAL4 and DcAL7 Expression Is Differentially Induced under Salt and ABA Treatments

To determine if *DcAL4* and *DcAL7* are induced by salt stress, we evaluated the relative transcript levels of both genes in eight-weeks old carrot plants under 250 mM acute salt treatment. Relative *DcAL4* transcript levels increase at 4, 6 and 8 h after salt treatment in leaves but not in roots and *DcAL7* transcript levels is induced in leaves at 6 h after salt treatment and in roots at 6 and 8 h after salt treatment (Figure 2d). As a control, we evaluated the relative transcript levels of *DcAREB3,* a carrot transcription factor that is known to be induced in carrot leaves and roots by salt stress [8]. These results provide evidence that both carrot *ALs* are upregulated under acute salt stress treatment, hinting at their role in abiotic stress responses. The differential pattern of expression in carrot leaves and roots also suggests that they may participate differentially in the salt stress response in vegetative tissues, as reported for other ALs [13]. 

ABA plays a key role in response to stresses such as heat, drought, salinity, high radiation and low temperature in plants [50]. In carrot, exogenous ABA modifies the expression level of *DcAREB3* (Appendix A; [8]) promoting the expression of ABA responsive genes. To determine if the induction of *DcAL4* and *DcAL7* under salt stress is mediated by ABA, we evaluated the relative transcript levels of both genes in 8 weeks-old carrot seedlings after ABA treatment, and in stressed plants supplemented with norflurazon (NFZ, an ABA biosynthesis inhibitor; [51]). It was observed that *DcAL4* relative transcript level, which is significantly induced in leaves by salt stress, is not induced when NFZ is included (Figure 2d) and that ABA treatment alone induces an increase in the transcript level of *DcAL4* in leaves at 4, 6 and 8 h (similar to NaCl treatments), and in roots after 8 h ABA treatment (Appendix A). In the case of *DcAL7*, the NFZ treatment also inhibits the induction observed in leaves and roots under NaCl treatment (Figure 2d), while ABA treatment results in a slightly increase in *DcAL7* transcript levels at 4 h in leaves and 8 h in roots, as found after NaCl treatment (Appendix A). Taken together, these data suggest that the NaCl-induced expression of *DcAL4* and *DcAL7* is promoted by ABA, and that *DcAL4* presents a greater expression in leaves under NaCl and ABA.

### 2.4. DcAL4 Heterologous Expression Generates an Increase in Total Carotenoids and Chlorophyll and Confers Enhanced Salt Tolerance in A. thaliana

To assess the functional role of both genes in vivo, multiple *DcAL4* and *DcAL7* T3 transgenic *A. thaliana* lines were generated (called Al4 and Al7, respectively; Appendix A). Expression analysis shows that all Al4 and Al7 transgenic lines present similar *DcAL4* and *DcAL7* transcript levels, without significative differences (Figure 3a). Three Al4 and Al7 representative transgenic lines were selected for further molecular and biochemical analysis.

In addition to PSY, 1-deoxy-D-xylulose-5-phosphate synthase (DXS) and 1-deoxy-D-xylulose-5-phosphate reductoisomerase (DXR) are key enzymes in terpenoid biosynthesis [52,53,54,55,56]. Both DXR and DXS have been proposed as control points in carotenoid and chlorophylls biosynthesis [57,58]. In this work, all Al4 transgenic lines analyzed presented an increase in *AtPSY* and *AtDXR*, but not in *AtDXS* transcript abundance (Figure 3b) suggesting a retrograde regulation given by DcAL4. In addition, an increase in total carotenoids and chlorophyll *a* and *b* compared to empty vector (EV) control lines was also observed (Figure 3c). On the contrary and although two Al7 transgenic lines present an increment in total carotenoids with respect to EV lines, there was no clear correlation between *AtPSY, AtDXS* or *AtDXR* expression, carotenoid content and chlorophyll *a* and *b* levels (Figure 3b,c). Therefore, we propose that *DcAL4* encodes for a transcription factor able to regulate positively the synthesis of carotenoids and chlorophylls in plants.

Next, transgenic Al4 and Al7 lines were selected for examining their performance under salt stress. Transgenic *A. thaliana* Al4 and Al7 plants watered with 200 mM NaCl solution for two weeks and then recovered for an additional two weeks, had a survival rate approaching 100%, while this parameter fell to 80% in EV *A. thaliana* control plants (Figure 3d). In addition, Al4 transgenic lines presented a reduction of only 30–40% in their fresh weight, while in EV and Al7 lines, fresh weight fell by over 70% (Figure 3d). Plant biomass was also affected after salt stress treatment; Al4 transgenic lines presented a lower reduction in biomass (20–40%), compared to Al7 and EV lines (60%). Concerning the plant height, even though only the Al4-L1, AL7-L2 and AL7-L3 lines were taller than EV plants in control conditions (well-watered plants), only Al4 lines were higher than EV plants after salt treatment (Figure 3d). Taking these results together, even though both *DcAL*s improve the survival rate, only the ectopic expression of *DcAL4* consistently raised the performance of the plants under salt treatment. These results suggest a functional discrepancy of both carrot ALs in plant responses to ABA and salt stress.

### 2.5. DcAL4 Heterologous Expression Generates an Increase in Total Carotenoids and Chlorophyll and Confers Enhanced Salt Tolerance in A. deliciosa

To assess the biotechnological projection of *DcAL*s in commercial plants, we overexpressed *DcAL4* and *DcAL7* in kiwi (*A. deliciosa* var Hayward). *A. deliciosa* is sensible to saline stress, showing a decrease in vegetative growth, accumulation of salts in the plant tissue, less development of the vegetative area and a lower photosynthetic yield under saline conditions [59,60]. Therefore, development of transgenic kiwi plants tolerant to salt stress could be a useful strategy to overcome this issue [61].

Kiwi is a dioecious plant with a long juvenile period of 4 to 5 years, it would take a lot of work and time to generate homozygous transgenic lines (15–20 years). Furthermore, considering that at commercial level kiwi plants are clonally propagated, in the present study we decided to work with the in vitro clonally propagated transgenic T0 lines. To reduce the chimerism level in the transgenic T0 kiwi lines, meristematic tissues were constantly subcultured in new propagation media with selection agent [62,63]. We obtained seven Al4 and eight Al7 lines that were selected by PCR (Appendix A). qRT-PCR analysis showed that all transgenic Al4 and Al7 kiwi lines analyzed present *DcAL4* and *DcAL7* expression (Figure 4a) although there were significative differences among each group. Three Al4 (L3, L6 and L9) and three Al7 (L1, L2 and L5) transgenic lines with similar transgene expression level and with the highest number of in vitro clones were selected in order to perform further analysis with all necessary replicates.

Al4 kiwi lines (L3, L6 and L9) showed an increase in total carotenoid, chlorophyll *a* and chlorophyll *b* contents, while Al7 kiwi lines (L1, L2 and L5) displayed a reduction in carotenoids and chlorophylls (Figure 4b). Subsequently, we carried out a qualitative and semi-quantitative analysis of the levels of hydrogen peroxide (H_2_O_2_) production in leaves subjected to an acute NaCl treatment by monitoring staining with DAB. Since high levels of salinity can trigger a highly oxidative cellular environment where the levels of reactive oxygen species (ROS) are increased (e.g., H_2_O_2_) [64,65]. DAB is oxidized by H_2_O_2_ in the presence of proteins containing the heme group (e.g., peroxidases), generating a brown precipitate that reflects the distribution of hydrogen peroxide in plant cells [66]. Leaves of Al4 kiwi lines subjected to an acute salt treatment (250 mM NaCl) showed a reduction in DAB staining compared to WT at 24 h of treatment, while leaves from Al7 lines did not show significant differences compared to WT leaves (Figure 4c and Appendix A).

As Al4 kiwi transgenic lines had increased carotenoid and chlorophylls levels and a reduction in ROS, the three lines (L3, L6 and L9) were subjected to chronic salt treatment in vitro (200 mM NaCl) to analyze the survival rate. In addition, one of the Al7 lines (L5) was picked randomly to observe their response to the stress treatment. During the treatments, transgenic plants showed a better and greener phenotype at 14 and 21 days after salt treatment (Figure 4d) and all Al4 lines showed an increased survival rate compared with the WT control (graph in Figure 4d). While all WT plants died at day 22, 50% of Al4 lines were still alive by day 30. Contrastingly, Al7-L5 presented only a slight increase in survival rate as around 85% died by day 22 (Figure 4d). Altogether, these results suggest that *DcAL4*, but not *DcAL7*, could be used as a biotechnological tool to enhance salt stress tolerance in *A. deliciosa*.

## 3. Discussion

In this study, we identified DcAL4 and DcAL7 proteins which have the conserved residues for H3K4me3/2 binding in their PHD domains (Figure 1a), suggesting that both DcALs could participate in chromatin remodeling. It has been proven that the valine at position 34, together with a hydrophilic amino acid, such as aspartic acid, glutamic acid and glutamine at position 35 are necessary for the PAL domain to bind to a G-rich motif [2]. By sequence alignment, we found that DcAL4 has a substitution at position 35(DQE→A) and DcAL7 at position 34(V→M) (Figure 1a). It is not clear what the effect of the substitution at position 35(DQE→A) in DcAL4 may be, but the substitution at position 34(V→M) in DcAL7 probably generate a loss of G-rich binding capability [2]. Even though the *A. thaliana* AtAL6 transcription factor does not bind to the G-rich motif due to a double mutation at amino acids 34(V→M) and 35(DQE→V) of the PAL domain [2], this does not result in a loss of function for AtAL6, since it is required for root hair development under phosphate starvation [10], it is key in the response to jasmonic acid in seedlings [67], and together with AtAL7, participates in seed germination under osmotic and salt stresses [11]. This suggests that DcAL4 and DcAL7 are functional and that the contrasting performance of ectopically expressed ALs in Arabidopsis and kiwi tolerance to salt stress arises from the molecular variation within the PAL and V domains, instead of the highly conserved PHD domain [2].

ALs can act as repressors or activators of gene expression. For example, the AtAL2 PAL domain interacts with structural components of Polycomb Repressive Complex 1 (PRC1) [3,11] which uses AtAL2 as a “reader” of the epigenetic marker H3K4me3, in order to generate the transition from an active to an inactive state of chromatin [11]. The mechanism by which ALs activate transcription has been described in much less detail. As an example, MsAL1 binds the promoter of the salt inducible gene *MsPRP2* in vitro [7] and the overexpression of *MsAL1* generates an increase in *MsPRP2* expression in alfalfa [12,68]. Similarly, we demonstrate that *DcAL4* and *DcAL7* encode functional transcription factors that bind to the promoter of *DcPSY2* and transactivate reporter genes in yeast (Figure 2b,c), suggesting that in carrot, both ALs could be regulating the expression of *DcPSY2*, previously reported to be induced by ABA [8]. 

In *A. thaliana*, the seven *ALs* are induced at the transcriptional level under saline stress conditions, and five respond to ABA treatment [2]. In *A. hortensis*, one of the four *ALs* (*AhAL1*) is upregulated under salt stress conditions, whereas *AhAL2*, *AhAL3* and *AhAL4* decrease their transcript levels in the same conditions [13]. It has been reported from the analysis of RNAseq data of different plants submitted to drought stress, that *ALs* are one of the most commonly up-regulated genes in seven different species [69]. Thereby, as for other *ALs*, *DcAL4* and *DcAL7* from *D. carota* also respond to ABA-mediated salt stress, with different expression patterns in roots and leaves (Figure 2d). Considering the transcriptional changes of *DcAL7* and *DcAL4* in roots and leaves in response to ABA treatment (Appendix A), it seems that ABA induces *DcAL4* gene expression in leaves earlier than in roots, an organ that has been suggested as the key source of dehydration sensing and ABA production under sustained stress-induced water deficit [70,71]. In addition, it has been reported that leaves and roots respond differentially at metabolic and transcriptional level to different types of abiotic stress [69,72,73,74,75]. Therefore, our results suggest an early leaves-specific ABA dependent function of *DcAL4* in *D. carota* under saline stress and a later induction in roots (Appendix A). A similar example can be shown in Shi et al. 2015 [76], were the *NtLCYB1* is significantly expressed in *Nicotiana tabacum* leaves after salt stress and drought and confers tolerance to these abiotic stresses when overexpressed in tobacco in correlation with an increment in carotenoid level [76]. 

One of the main physiological processes affected by stress conditions in plants is the photosynthesis [77]. It has been reported that an increase in chlorophylls and carotenoids improves the photosynthetic capacity and fitness of plants [26,27,35]. In addition, plants with higher antioxidant content can eliminate ROS produced during abiotic stress protecting its cells from oxidative damage [14,16,78]. DXR, a key enzyme in the synthesis of metabolic precursors of chlorophylls and carotenoids, plays a significant role in the regulation of stress conditions and participates in multiple physiological responses [53,79]. Likewise, PSY is a key player and the bottleneck in carotenoid and ABA synthesis in response to abiotic stress conditions [8,80]. In this work, we observed that the ectopic expression of *DcAL4* but not *DcAL7* in *A. thaliana* induces the expression of *AtDXR* and *AtPSY* (Figure 3b), which correlate with the increased chlorophylls and carotenoid content in *A. thaliana* leaves (Figure 3c). Similarly, only *DcAL4 A. deliciosa* transgenic lines show an increase in carotenoids and chlorophylls. This increase in carotenoids together with chlorophyll *a* and *b* in Al4 transgenic lines suggest an improvement in the antioxidant and photosynthetic capacity of these plants, which in turns help them to cope with the harsher environmental conditions. On the contrary, *DcAL7 A. deliciosa* transgenic lines did not overcome ROS (Figure 4c), which could be also associated with less carotenoids and chlorophylls (Figure 4b). Furthermore, *DcAL7* may have other roles, but we discard those associated with macroscopic development since we did not notice any morphological difference in roots (length) and aerial parts of Arabidopsis and kiwis overexpressing this transcription factor.

Due to the role of ALs in abiotic stress tolerance, ectopic expression of *ALs*, such as *MsAL1* from alfalfa [9,12], *GmPHD2* from *Glycine max* (soybean) [81], *AhAL1* from *A. hortensis* [13] and *AtAL5* from *A. thaliana* [2] have been proposed as biotechnological strategy to increase tolerance to saline and water stress. In this sense, we observed that *A. thaliana* and *A. deliciosa* transgenic lines for *DcAL4* have a better performance (survival rate, fresh weight, biomass, oxidative damage) after saline stress treatments (Figure 3d and Figure 4c,d) compared to control plants, while *DcAL7* lines only present an increase in survival rate, but no an overall improvement in plant fitness under salt stress. In addition, it is important to mention that no pleiotropic effects were observed in the *A. thalian*a and *A. deliciosa* lines overexpressing *DcAL4* and *DcAL7* similar than in other transgenics plants overexpressing ALs [2,13], in contrast with the overexpression of AREB/ABF transcription factors also involved in abiotic stress tolerance response [82,83]. Preliminary results showed that the overexpression of *DcAREB3* in *A. thaliana* produces plant with low and delayed germination while in *A. deliciosa*, it produces transgenic plants with leaf deformity and low grow rate (Data not shown). Future work could also focus on determining if DcAL4 and DcAREB3 transcription factors interact in vivo and regulate coordinately the expression of *DcPSY2* in leaves under abiotic stress. 

Considering the results presented, we update the model of *DcPSY2* regulation under abiotic stress proposed in Simpson et al., 2018 (Figure 5), suggesting that *DcAL4* encodes for a functional transcription that regulates the carotenoids/ABA biosynthesis under saline stress in carrot leaves. We propose that *DcAL4* could be a prime candidate for improving the performance and fitness of commercial crops under saline conditions, thus enabling the use of soils that are currently not suitable for cultivation.

## 4. Materials and Methods

### 4.1. Plant Material

Wild-type (Col-0) and transgenic *A. thaliana* seeds were surface sterilized by agitating them in a solution containing 95% ethanol for 1 min and 2.62% (*v*/*v*) sodium hypochlorite for 15 min. Following four washes in sterile water and dried on sterile paper. The sterilized seeds were placed in plates with solid half MS medium ([84]; 2.2 g/L MS salts, 1% sucrose, 0.01% myo-inositol, 0.22% vitamins and 0.7% agar pH 5.7 with or without antibiotic), and maintained at 22 °C in a growth chamber with a 16 h long day photoperiod (white fluorescent light; 115 μmol m^−2^ s^−1^) for 4 weeks. Transgenic lines were subsequently moved to a greenhouse and cultivated in a soil:vermiculite (2:1) for all further analyses. Seedlings of carrot (*Daucus carota* cv. Nantesa, 4 weeks old) were cultivated in a greenhouse (16 h long day photoperiod at 22 °C) in soil:vermiculite (2:1) and subjected to salt stress and molecular analyses. All plants cultivated in the greenhouse were watered with a standard hydroponic medium (0.125 mM KNO_3_, 0.15 mM Ca(NO_3_)_2_·4H_2_O, 0.075 mM MgSO_4_·7H_2_O, 0.05 mM KH_2_PO_4_, 5 μM KCl, 5 μM H_3_BO_3_, 1 μM MnSO_4_, 200 nM ZnSO_4_·7H_2_O, 150 nM CuSO_4_, 10 μM Na_2_O_3_Si, 10 μM Fe/DTPA, pH 6). Kiwi plants (*Actinidia deliciosa* cv. Hayward, CA, USA) were a generous gift of Viverosur (https://viverosur.com/, accessed on 12 January 2022). New shoots were surface sterilized using 70% ethanol (30 s) and incubation in 10% (*v*/*v*) sodium hypochlorite and 1 drop of Tween 20 (20 min). Shoots were then washed 4 times with sterile water and placed in flasks for in vitro culture that contained solid half strength MS medium ([84]; 2.2 g/L MS salts, 3% sucrose, 0.22% vitamins, 0.01% myo-inositol and 0.7% agar pH 5.7), supplemented with 0.5 mg/L BAP.

### 4.2. Vector Construction

To determine subcellular localization and for expressing DcAL4 and DcAL7 in plants, their CDS without the stop codon were cloned into pCR^®^8/GW/TOPO (Invitrogen). Positive clones were verified by PCR, restriction assay and sequencing (Macrogen Corp. Rockville, USA). Next, pCR8/Dc*ALs* were recombined into pk7RWG2 (pK) forming the pKAL4 and pKAL7 binary vectors for the expression of DcALs:RFP fusion proteins. All Clones were evaluated by both PCR and restriction assay, before transforming into GV3101 *Agrobacterium tumefaciens* strain. For monohybrid and transactivation assays, the *DcPSY2* promoter region (798 bp) was cloned upstream of the gene encoding AurobasidinA (*AbA*) resistance in the pP2-pAbAi vector. Subsequently, by recombining the pCR8*/*Dc*ALs* plasmids into pDEST22 and pDEST32 (Invitrogen) the pDEST22/Dc*ALs* (Dc*ALs* fused to the GAL4 activation domain) and pDEST32/Dc*ALs* (Dc*ALs* fused to the GAL4 DNA binding domain) plasmids were generated. In a previous study, CAREB1:RFP was constructed [8].

### 4.3. Agrobacterium Mediated Transformation of A. thaliana and A. deliciosa

*A. thaliana* (Col-0) was transformed with *A. tumefaciens* containing pKAL4, pKAL7 or pK (empty vector, EV) by floral dip method [85]. T1 seeds were selected on 100 mg/L kanamycin, transported to a greenhouse and evaluated by PCR to determine the presence of the *DcAL4* or *Dc**AL7* transgene (see below). To select for homozygous lines, 12 T2 kanamycin-resistant plants per T1 line were chosen and propagated in greenhouse conditions until seed production. Homozygous T2 lines were selected when all T3 seeds were resistant to kanamycin. T3 homozygous lines were employed for subsequent molecular and functional evaluations. 

*A. deliciosa* was transformed using a modified protocol based on Kim [86]. Explants from petioles and leaves (~1 cm) cultivated in vitro were submerged and kept in solution with *A. tumefaciens* transformed with pK7AL4 or pKAL7 for 30 min while the explants were scraped using a scalpel. After drying using sterile paper, explants were positioned on plates with solid half strength MS medium (2.2 g/L MS salts, 3% sucrose, 0.22% vitamins, 0.01% myo-inositol and 0.7% agar pH 5.7) in complete darkness at 24 ± 2 °C for about 48 h. Subsequently, explants were washed (3–5 times) with sterile water, dried using sterile paper, and maintained in medium I for somatic organogenesis until transforming shoots were regenerated (Appendix A). On production of shoots (week 15–20), these were moved to half strength MS for elongation and overcrowding.

### 4.4. Chronic Salt Treatments of A. thaliana and A. deliciosa

Transgenic *DcAL4*, *DcAL7* and EV T3 *A. thaliana* lines were transported from in vitro to greenhouse conditions and acclimated for 2 weeks. To impose salt stress, pots were watered for 14 days with hydroponic medium containing 200 mM NaCl, then allowed to recover for a further 14 days under normal irrigation (hydroponic medium lacking NaCl). On completion of this treatment, survival rate, fresh weight, biomass and plant height were measured, and the plants imaged. All experiments were undertaken in triplicate (3 pots) with 9 plants of each transgenic T3 line in each pot.

For *A. deliciosa* salt treatment, wild-type and 6 transgenic Al4 lines were maintained in MS medium containing 200 mM NaCl for 6 weeks and cultivated at 22 °C ± 2 °C and 16 h photoperiod. Six plants of each transgenic line and 12 WT plants (two replicates of WT at two different times) were used in this assay. The plants were observed every 48 h and the survival percentage was measured. Plants were classified as dead on presenting a brown phenotype. Moreover, images were captured to compare the phenotypes of the plants. 

### 4.5. Acute Salt Stress, ABA and Norflurazon Treatments of Carrot

Carrot plants (8 weeks) were immersed in 100 μM ABA or 250 mM NaCl supplemented or not with 10 μM norflurazon (NFZ) for 2, 4, 6 and 8 h. Following these treatments, tissue from the aerial part (leaves) and from the carrot root was sampled. The assay was performed with 3 biological replicates for each sampling time (3 plants per sample), each with 2 technical replicates. Total RNA was then isolated for gene expression analysis.

### 4.6. Acute Salt Stress and DAB Staining in A. deliciosa Leaves

*A. deliciosa* leaves from WT and transgenic lines were submerged in 250 mM NaCl for 0 and 24 h. At both timepoints, 3 leaves from each line were chosen; a control treatment (no NaCl) was also undertaken at each timepoint. The assay was performed twice. Hydrogen peroxide levels were quantified using 3,3′-Diaminobenzidine (DAB) staining, following the modified protocol of Daudi and O’Brien [66]. Treated leaves were submerged in DAB staining solution (1 mg/mL DAB, 10 mM Na_2_HPO_4_ and 0.05% Tween 20), a gentle vacuum for 5 min was applied and then left in darkness and constant shaking for 4 h. After completion of this incubation, the staining solution was discarded, and leaves were submerged in the bleaching solution (Ethanol: Glycerol: Acetic acid = 3:1:1) and placed in a thermoregulated bath (80 °C, 20 min). Subsequently, the bleaching solution was refreshed, and samples were returned to the bath for a further 30 min. On completion of the second incubation, samples were dried on absorbent paper and the degree of DAB staining was photographed. To plot DAB staining levels, a false-colored image was made using ImageJ, and the percentage of the area of each leaf corresponding to pixels 170–255 were counted (Appendix A).

### 4.7. RNA Extraction and Quantitative RT-PCR (qRT-PCR)

Frozen powder of *A. thaliana* leaves, *A. deliciosa* leaves, and carrot (leaves and roots) was used for extraction of total RNA with CTAB Buffer (2% CTAB, 2 M NaCl, 25 mM EDTA, 100 mM TrisHCl (pH 8.0), 2% PVP40 (PM. 40,000), 0.05% Trihydrochlorate of spermidine and 2% β-mercaptoethanol) method [87]. Traces of genomic DNA were removed by DNaseI treatment for 20 min. In order to synthesize cDNA, 3 μg of treated RNA was mixed with 1 mM oligo AP primer (Appendix A) and ImpromII reverse transcriptase (Promega). For qRT-PCR reactions it was used the LightCycler system (Stratagene), employing SYBR Green double strand DNA binding dye [31]. Specific primers were designed to target *DcAL4, DcAL7, DcPSY2*, *DcAREB3, DcUbiquitin, AtPSY, AtUbiquitin* and *Ad18S* (Appendix A). *AtUbiquitin, DcUbiquitin* and *Ad18S* were employed as housekeeping for *A. thaliana, D. carota* and *A. deliciosa,* respectively. The relative transcript levels of these genes in the different treatments examined in this research were determined using the crossing point values and previously described equations [88]. All qRT-PCR reactions were undertaken using 3 biological and 2 technical replicates. Reaction specificity was examined by gel electrophoresis and melting gradient dissociation curves for all genes.

### 4.8. Pigment Quantification

*A. thaliana* and *A. deliciosa* leaves (100 mg) were extracted to obtain pigments using 1 mL hexane/ethanol/ acetone (2:1:1 *v*/*v*) as previously described [31]. Leaf blanching was achieved after 2 successive rounds of extraction. After drying extracts with gaseous nitrogen and resuspension in 2 mL acetone, pigments were quantified in triplicate by a spectrophotometer at 474, 645, 662 and 750 nm to measure carotenoids, chlorophyll *b*, chlorophyll *a*, and the turbidity of the sample, respectively [89]. All samples were maintained in dark and ice to prevent isomerization, photo-degradation and/or others structural alterations of the extracted pigments.

### 4.9. Subcellular Localization

The pK/AL4, pK/AL7 and pCAMBIA1302 (CaMV35S::GFP, Marker Gene Technologies, Inc.) plasmids were used for the transient expression in *Nicotiana tabacum* (tobacco) leaves by agroinfiltration [90]. The leaf samples were then observed using an inverted epifluorescence microscope (IX-70, Olympus America Inc., Melville, NY). GFP and RFP fluorescence images were captured using excitation at 450–490 nm (blue light) and at 530–560 nm (green light), respectively, with 40× augmentation. All images were subsequently analyzed and processed with LSM5 Image Browser and Adobe Photoshop software.

### 4.10. Bioinformatic Analysis of the DcPSY2 Promoter

ALs binding motifs were detected using MotifScanner (http://homes.esat.kuleuven.be/~sistawww/bioi/thijs/download.html, accessed on 12 January 2022). For PWM (Position Weight Matrices), the PlantCARE Database (http://bioinformatics.psb.ugent.be/webtools/plantcare/html/, accessed on 12 Januray 2022) was employed, as it harbors 435 regulatory motifs from mono and dicotyledonous plants. As the *D. carota* genome sequence was not available at the start of our research, the *A. thaliana* genome was used as the background model, the only third-order plant organism within the candidate list, as recommended (http://toucan.aertslab.org/software/toucan.php#man, accessed on 12 January 2022). A “prior” (probability of finding a particular motif) value of 0.7 was set and only the positive strand of DNA was scanned. All transcription initiation sites (TSSs) were determined by the BDGP bioinformatic program (http://www.fruitfly.org/seq_tools/other.html, accessed on 12 January 2022).

### 4.11. Monohybrid and Transactivation Assay

A monohybrid assay was carried out as previously described [8] using Matchmaker^®^ Gold Yeast One-Hybrid Library Screening System (Clontech). The yeast Y1HGold strain transformed with pP2-AbAi, in which the *DcPSY2* promoter was cloned upstream of the *aureobasidin A* (*AbAi*) resistance gene, was employed. The manufacturer’s protocol was used to measure the minimum inhibitory concentration of AbAi. To assay the union of ALs transcription factors to the *DcPSY2* promoter, the pDEST22/ALs vectors were transformed into the Y1HGold-pP2-AbAi strain and cultivated in SD/-Trp-Ura medium. The binding of ALs to the *DcPSY2* promoter was determined as the capacity of the various yeast strains to grow in the presence of AbAi. For transactivation assays, the MaV203 yeast strain was transformed with the pDEST32/ALs plasmids [49], and cultivated in SD/-Leu medium. As the MaV203 strain harbors stably integrated GAL4-inducible *HIS3* and *URA3* reporters, the transactivation of these reporter genes was determined as the ability of the yeast strains to thrive without both uracil (SD/-Ura) and histidine (SD/-His + 25 mM/50 mM 3AT, a competitive inhibitor of HIS3, meaning that only baits that exhibit self-activation grow in the presence of 3-AT).

## Figures and Tables

**Figure 1 ijms-23-12157-f001:**
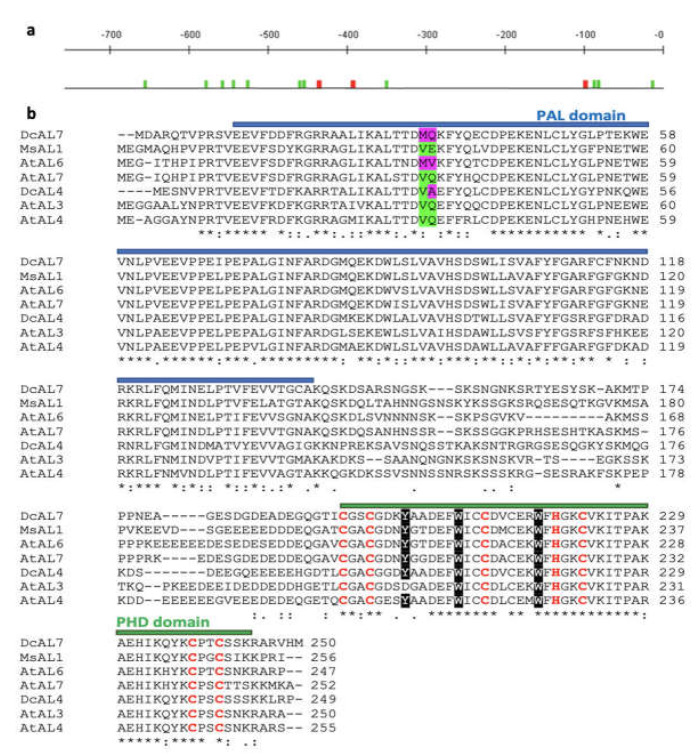
Alignment of plant ALs and ALFIN regulatory elements in the *DcPSY2* promoter. (**a**) The 769 bp promoter *DcPSY2* [8] presents 11 ALFIN (green) and three ABRE (red) responsive elements. (**b**) Sequence alignment of DcAL4 and DcAL7 from *D. carota*, MsAL1 (GenBank: AAA20093.2) from *M. sativa*, and AtAL3 (Gene ID: 823316), AtALAL4 (Gene ID: 832690), AtALAL6 (Gene ID: 814776) and AtAL7 (Gene ID: 838013) form *A. thaliana*. The PAL domain is highlighted in blue and the PHD domain in green. Conserved amino acids within the PHD domain are highlighted in red (Cys4HisCys3) and black (Tyr-Trp2). Variations in the capability to bind G-Rich motifs in the PAL domain are highlighted in green and purple. * = Identical amino acids; : = conserved; . = semi-conserved, respectively.

**Figure 2 ijms-23-12157-f002:**
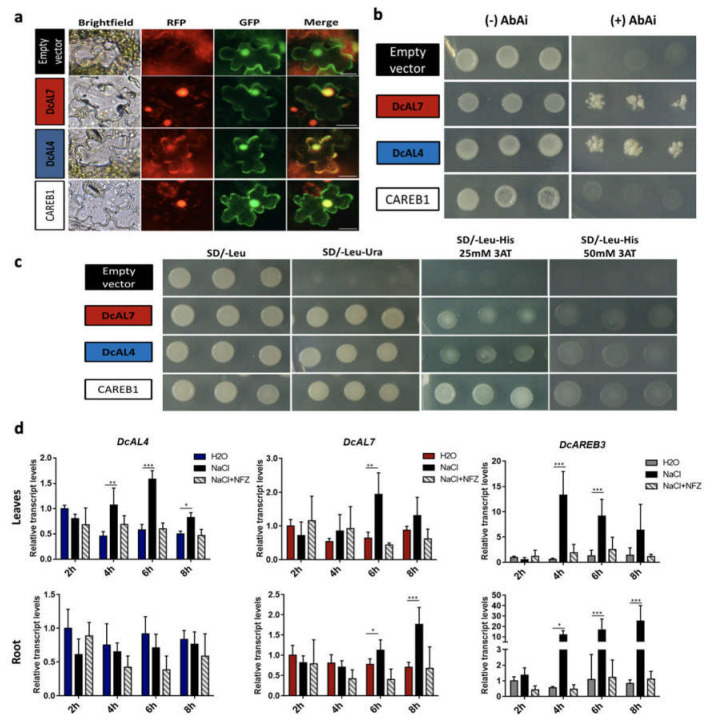
Functional characterization of *DcAL4* and *DcAL7*. (**a**) Subcellular localization of DcAL4 and DcAL7. Co-infiltration of tobacco leaves with Agrobacterium carrying the DcAL4:RFP and DcAL7:RFP plasmids, 35S:GFP (cytoplasmic and nuclear marker) or DcCAREB1:RFP (nuclear localization positive control; GenBank: EU433292.1) vectors. RFP red channel: Images taken using the Cy3 filter. GFP Green channel: Images taken using the FITC filter. Bar: 25 μM. (**b**) Monohybrid assay to determine the DNA binding capability of DcAL4 and DcAL7 transcription factors to the *DcPSY2* promoter. The strains were grown on SD/-Ura-Trp medium supplemented with or without Aurobasidin A (AbAi). (**c**) Transactivation assay using DcAL4 and DcAL7 transcription factors to induce the expression of the reporter genes *URA3* and *HIS3* in the reporter yeast strain grown on SD (SD/-Leu medium, SD baseline without leucine), SD/-Ura (auxotrophy for uracil), SD/-His + 25 mM and 50 mM 3AT (auxotrophy for histidine). (**d**) *DcAL4* and *DcAL7* relative transcript levels in *D. carota* leaves and roots under acute salt (250 mM NaCl) and ABA synthesis inhibitor (norflurazon, NaCl-inh) treatments. *DcAREB3* transcript levels were used as treatment control gene. Transcript abundance was normalized to *DcUbiquitin* transcript levels and the H_2_O^−^ 2 h control condition was used as calibrator. All values represent the means of three independent replicates (+SD). Statistically significant differences were determined by two-tailed ANOVA test: * *p* < 0.05, ** *p* < 0.01, *** *p* < 0.001.

**Figure 3 ijms-23-12157-f003:**
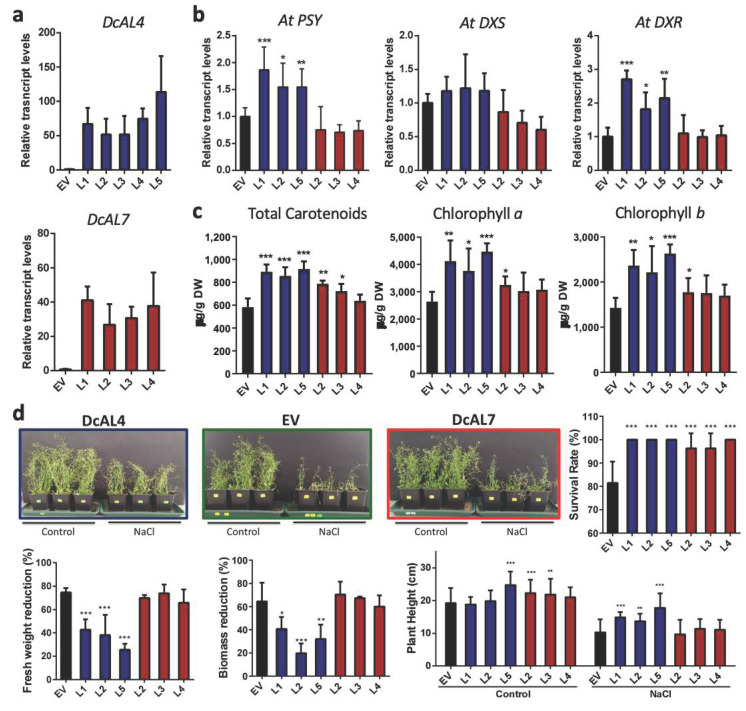
Effect of DcAL4 and DcAL7 ectopic expression in *A. thaliana*. (**a**) *DcAL4* and *DcAL7* relative transcript level in *A. thaliana* Al4 and Al7 T3 lines, respectively. (**b**) *AtPSY*, *AtDXS* and *AtDXR* relative transcript level in selected Al4 and Al7 T3 lines. (**c**) Carotenoid and Chlorophylls content in leaves of selected Al4 and Al7 lines. (**d**) Salt stress tolerance of *A. thaliana* transformed with *DcAL4* and *DcAL7.* Representative image of EV, Al4 and Al7 lines subjected to salt stress treatment (see Appendix A for details of each line). Survival rate, fresh weight, biomass and plant height of EV, Al4 and Al7 lines submitted to salt stress treatment. For (**a**,**b**), transcript abundance was normalized to *At**Ubiquitin* relative transcript levels and the Empty vector (EV) was used as calibrator. All values represent the means of three independent replicates (+SD). Statistically significant differences were determined by one-tailed ANOVA test (relative transcript levels and salt treatment) and by two-tailed unpaired T test (chlorophylls and carotenoids): * *p* < 0.05, ** *p* < 0.01, *** *p* < 0.001. Control: Well-watered plants; NaCl: 200 mM NaCl treatment.

**Figure 4 ijms-23-12157-f004:**
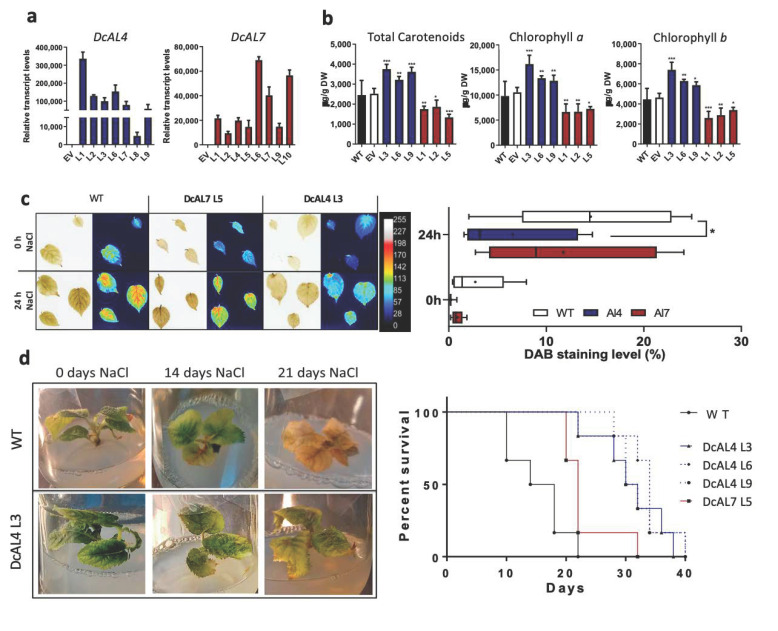
Effect of DcAL4 and DcAL7 ectopic expression in *A. deliciosa.* (**A**) *DcAL4* and *DcAL7* relative transcript levels in *A. deliciosa* Al4 and Al7 T0 lines, respectively. (**B**) Carotenoid and Chlorophylls content in leaves of selected Al4 and Al7 lines. (**C**) Hydrogen peroxide content in leaves of Al4 and Al7 *A. deliciosa* lines subjected to acute treatment with NaCl. Representative image of WT, Al4 and Al7 *A. deliciosa* leaves pre- and post-treatment with 250 mM NaCl stained with DAB, as well as a false-colored image generated by ImageJ software (see Appendix A for details of each line). The graph shows the quantification of the most oxidized area, considering the percentage of the area in the leaves covered with pixel intensities between 170 to 255 in the false-colored image. Each bar represents the average of all Al4 and Al7 lines used in this assay. (**D**) In vitro chronic salt stress treatment in Al4 *A. deliciosa* transgenic lines. The images show representative WT and Al4-L3 plants at 0, 14 and 21 days under salt treatment. The graph shows the percent survival of transgenic and control lines in the time. For (**A**) transcript abundance was normalized to *Ad18S* transcript levels. All values represent the means of three independent biological replicates (+SD). Statistically significant differences were determined by one-tailed ANOVA test (relative transcript levels), by two-tailed unpaired T test (chlorophylls and carotenoids) and by two-tailed ANOVA test (DAB assay): * *p* < 0.05, ** *p* < 0.01, *** *p* < 0.001.

**Figure 5 ijms-23-12157-f005:**
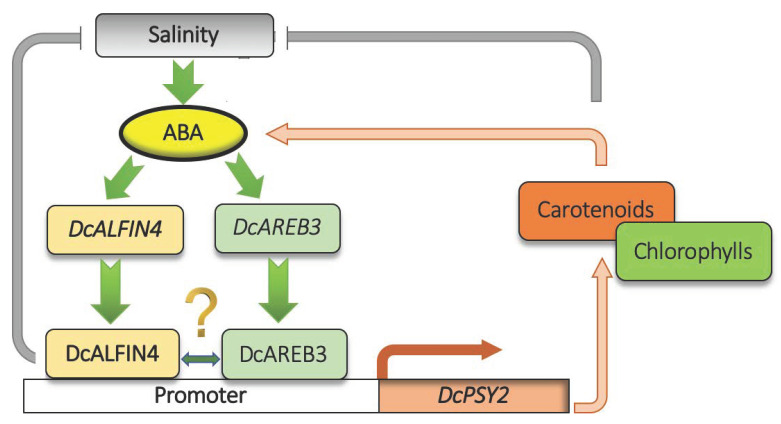
Proposed model for the role of DcAL4 in the induction of *DcPSY2* expression during salt stress in leaves. Salt stress and ABA produce an increment in the expression of *DcPSY2* and *DcAREB3* in carrot leaves (This study and [8]), and *DcAL4* in leaves (this study). The expression of *DcPSY2* is induced by ABA as a positive feedback mechanism accomplished by the direct binding of DcAL4 (this study) and DcAREB3 [8] transcription factors. The upregulation of *DcPSY2* leads to an increment in carotenoids and chlorophylls, which together promote salt stress tolerance responses.

## Data Availability

Stange, Claudia et al. (2022), Appendix A, Dryad, Dataset, https://doi.org/10.5061/dryad.866t1g1sx, accessed on 12 January 2022.

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
