# Peer review of "Carrot DcALFIN4 and DcALFIN7 Transcription Factors Boost Carotenoid Levels and Participate Differentially in Salt Stress Tolerance When Expressed in Arabidopsis thaliana and Actinidia deliciosa"

_ijms, 2022, doi:10.3390/ijms232012157_

Round 1
Reviewer 1 Report
The authors mainly explore the function of carrot AL4 and AL7 in carotenoids and salt stress, transgenic results provide a very important evidence, but some of them should be approved.
1. The introduction was too short, carotenoid related research in carrot or other plants could be added.
2. The expression level of PSY, DXS and DXR were measured in Arabidopsis, add these in kiwi.
3. The expression level of AL4 was significantly higher in Nacl treatment in D. carota leaves, while no difference in Root, Whether they really participate in salt stress?
4. AL4 might involve in carotenoid and chlorophylls synthesis based on transgenic arabidopsis and Y1H results, LUC and transgenic kiwi results should be added to approve the results.
5. The contents of carotenoids were no significant difference in transgenic AL7 Arabidopsis, while significant decreased in kiwi line, could you discuss the results?
6. For transgenic lines, such as AL7 Arabidopsis, why select L2/3/4 lines, why not L1? In AL4 kiwi, why L3/6/9, why not L1? the expression level of AL4 was more higher than that in other line.
Author Response
Q1. The introduction was too short, carotenoid related research in carrot or other plants could be added.
R1: Dear reviewer, we include more information regarding carotenoids and abiotic stress response in carrot in the introduction
Q2. The expression level of PSY, DXS and DXR were measured in Arabidopsis, add these in kiwi.
R2: Dear reviewer, it's a good suggestion. Indeed, we explore this possibility. Unfortunately, Actinidia deliciosa is a hexaploidy variety and makes it very risky to select some carotenogenic genes because it presents several paralogs of carotenogenic genes. For instance, in the diploid Actinidia chinensis variety whose genome is sequenced, we found 7 putative DXS, 2 putative DXR and 4 putative PSY genes. This scenario makes it very uncertain to select some paralogs for each DXS, DXR and PSYs or to analyze the expression of all the families together. In any case, the results obtained could be only an approximation to answer about the increment in carotenoids in the kiwi AL4 transgenic lines. A similar example was shown by Shi et al. 2015 [76], were the overexpression of NtLCYB1 in Nicotiana tabacum confers tolerance to salt stress and drought in correlation with an increment in carotenoid level.
Q3. The expression level of AL4 was significantly higher in Nacl treatment in D. carota leaves, while no difference in Root, Whether they really participate in salt stress?
R3: Dear reviewer, as you mentioned, although the expression of AL4 was significantly not different in carrot roots under salt treatment, DcAL4 expression is induced in roots and leaves in response to ABA treatment (Figure S2). ABA seems to induce DcAL4 gene expression in leaves earlier than in roots, an organ that has been suggested as a key source of dehydration sensing and ABA production under stress-induced water deficit [61, 62]. Also, it has been reported that leaves and roots respond differentially at metabolic and transcriptional level to different types of abiotic stress [60, 63-66]. Therefore, our results suggest an early leaves-specific ABA dependent function of DcAL4 in D. carota under saline stress and a later induction in roots (Figure S2). A similar example can be shown in Shi et al 2015, were the NtLCYB1 is significantly expressed in leaves after salt stress and drought and confers tolerance to these abiotic stresses when overexpressed in tobacco. Considering this analysis, we can conclude that Arabidopsis and kiwi Al4 OE plants present a higher tolerance to salt stress evidenced by the functional analysis.
Q4. AL4 might involve in carotenoid and chlorophylls synthesis based on transgenic arabidopsis and Y1H results, LUC and transgenic kiwi results should be added to approve the results.
R4: The overexpression of AL4 in both different species, Arabidopsis and kiwi, produces a significant increment in carotenoids and chlorophyll a and b (Figures 3 and 4). Similar results were obtained after overexpressing carotenogenic genes in different plants (Moreno et al., 2013), but most important is that the overexpression of carotenogenic genes can lead to abiotic stress tolerance and carotenoid increment (Kang et al., 2018; Shi et al., 2015). Therefore, we suggest AL4 is involved in carotenoid and chlorophylls synthesis. This conclusion is complemented by the fact that AL4 is capable of binding to the DcPSY2 promoter and to induce the expression of carotenogenic genes in Arabidopsis.
Q5. The contents of carotenoids were no significant difference in transgenic AL7 Arabidopsis, while significant decreased in kiwi line, could you discuss the results?
R5: It is a good observation. AL7 produces in Arabidopsis a significant increment in carotenoids in two lines but not in chlorophylls, but unexpectedly in kiwi the AL7 OE lines present a significant reduction in total carotenoids and chlorophylls. These results could be related to the endogenous regulation of carotenoids and chlorophylls synthesis and degradation in both species. Indeed, AL7 did not produce tolerance to salt stress measured as survival rate and in H2O2 reduction (determined by DAB), which could be also associated with less carotenoids and chlorophylls (Figure 4B-C). It is interesting but it is a phenomenon that is repeated when expressing genes in plants. As an example, when DcLCYB1 was expressed in tobacco it produces higher increment in carotenoids than in carrot (Moreno et al 2013; 2016).
Q6. For transgenic lines, such as AL7 Arabidopsis, why select L2/3/4 lines, why not L1? In AL4 kiwi, why L3/6/9, why not L1? the expression level of AL4 was more higher than that in other line.
R6: Dear reviewer, regarding the Arabidopsis AL7 transgenic lines, all lines showed similar expression levels (between 30-40 fold, Figure 3a). We selected L2, L3 and L4 because they presented only one insertional event by the Mendelian segregation of the resistance gene. Additionally, L1 did not show conclusive results, in one of the stress replicates it showed higher tolerance compared with control, while in a next replicate it did not, and therefore we cannot discard an insertional effect. Regarding the AL4 transgenic line in kiwi, we select lines with similar expression levels and sufficient leaves and tissue for all the analysis shown in Figure 4. Kiwi plants have to be clonally propagated in vitro, and due to the restriction related to the pandemic we lost some of the lines. We performed the in vitro salt stress treatment with the remaining Al4 lines (L1, L2, L3, L6 and L9), all AL4 lines showed better performance than controls (Of note, L1 was not better than other Al4 lines), but only L3, L6 and L9 had enough replicates for all the experiments that were presented.
Reviewer 2 Report
This manuscript by Quiroz et al. describes the isolation and characterization of two putative Als encoding genes from carrots. The authors found that DcAL4 binds to carrot PSY2, consequently increasing carotenoid biosynthesis and likely activating plant tolerance to salt. AL4 overexpression effects were validated in two important crops (carrot and kiwi), which is impressive.
While the authors clearly stated changes in carotenoids, chlorophyll, and tolerance, it was not clear whether untreated transgenic plants are indistinguishable from the non-transgenic controls in terms of plant architecture (i.e., height, root architecture, leaf shape, etc.). If possible, I would suggest adding this information to the paper.
Overall, I think this is a well-written manuscript with results that support the conclusions.
The findings presented are attractive, well documented, and fit well with the journal’s scope.
Author Response
Q1: This manuscript by Quiroz et al. describes the isolation and characterization of two putative Als encoding genes from carrots. The authors found that DcAL4 binds to carrot PSY2, consequently increasing carotenoid biosynthesis and likely activating plant tolerance to salt. AL4 overexpression effects were validated in two important crops (carrot and kiwi), which is impressive.
R1: Dear reviewer, we appreciate the positive comment
Q2: While the authors clearly stated changes in carotenoids, chlorophyll, and tolerance, it was not clear whether untreated transgenic plants are indistinguishable from the non-transgenic controls in terms of plant architecture (i.e., height, root architecture, leaf shape, etc.). If possible, I would suggest adding this information to the paper.
R2: It's a very interesting comment. We didn’t see any pleiotropic effects in the A. thaliana and A. deliciosa lines overexpressing DcAL4 or DcAL7, similar than has been reported for other transgenics plants overexpressing Als [2, 12]. We include this comment in the discussion section.
Q3: Overall, I think this is a well-written manuscript with results that support the conclusions. The findings presented are attractive, well documented, and fit well with the journal’s scope.
R3: Thank you, we appreciate the positive comment
Round 2
Reviewer 1 Report
The manuscript can be accepted in present form.